# Toll-like Receptor 2 as a Marker Molecule of Advanced Ovarian Cancer

**DOI:** 10.3390/biom11081205

**Published:** 2021-08-13

**Authors:** Małgorzata Sobstyl, Paulina Niedźwiedzka-Rystwej, Rafał Hrynkiewicz, Dominika Bębnowska, Izabela Korona-Głowniak, Marcin Pasiarski, Barbara Sosnowska-Pasiarska, Jolanta Smok-Kalwat, Stanisław Góźdź, Anna Sobstyl, Wojciech Polkowski, Jacek Roliński, Ewelina Grywalska

**Affiliations:** 1Department of Gynecology and Gynecological Endocrinology, Medical University of Lublin, 20-037 Lublin, Poland; malgorzata.sobstyl@umlub.pl; 2Institute of Biology, University of Szczecin, 71-412 Szczecin, Poland; dominika.bebnowska@usz.edu.pl; 3Department of Pharmaceutical Microbiology, Medical University of Lublin, 20-093 Lublin, Poland; iza.glowniak@umlub.pl; 4Department of Immunology, Faculty of Health Sciences, Jan Kochanowski University, 25-317 Kielce, Poland; marcinpasiarski@gmail.com; 5Department of Hematology, Holy Cross Cancer Centre, 25-734 Kielce, Poland; 6Department of Oncocardiology, Holy Cross Cancer Centre, 25-734 Kielce, Poland; spbasia@gmail.com; 7Department of Clinical Oncology, Holy Cross Cancer Centre, 25-734 Kielce, Poland; jolantasmok1@gmail.com (J.S.-K.); stanislawgozdz1@gmail.com (S.G.); 8Faculty of Medicine and Health Sciences, The Jan Kochanowski University, 25-516 Kielce, Poland; 9Department of Clinical Immunology and Immunotherapy, Medical University of Lublin, 20-093 Lublin, Poland; sobma@poczta.onet.pl (A.S.); jacek.rolinski@gmail.com (J.R.); ewelina.grywalska@gmail.com (E.G.); 10Department of Surgical Oncology, Medical University of Lublin, Radziwiłłowska 13 St., 20-080 Lublin, Poland; wojciech.polkowski@umlub.pl

**Keywords:** Toll-like receptor, TLR2, ovarian cancer, biomarker, receptor, expression

## Abstract

Ovarian cancer is a global problem that affects women of all ages. Due to the lack of effective screening tests and the usually asymptomatic course of the disease in the early stages, the diagnosis is too late, with the result that less than half of the patients diagnosed with ovarian cancer (OC) survive more than five years after their diagnosis. In this study, we examined the expression of TLR2 in the peripheral blood of 50 previously untreated patients with newly diagnosed OC at various stages of the disease using flow cytometry. The studies aimed at demonstrating the usefulness of TLR2 as a biomarker in the advanced stage of ovarian cancer. In this study, we showed that TLR2 expression levels were significantly higher in women with more advanced OC than in women in the control group. Our research sheds light on the prognostic potential of TLR2 in developing new diagnostic approaches and thus in increasing survival in patients with confirmed ovarian cancer.

## 1. Introduction

Ovarian cancer (OC) is considered the second most common cause of death from gynecological cancer in women after cervical cancer.

Ovarian cancer (OC) is considered the eighth most common cause of death from cancer in women in the world, in addition to being one of the most common gynecologic cancers, which occupies third place in mortality due to gynecological cancer in women after cervical cancer and uterine cancer [1].

About 295,414 new cases of this cancer are diagnosed annually (6.6% of all cancer cases), and it is estimated that it is responsible for approximately 184,799 deaths per year (3.9% of all cancer deaths). The highest incidence and death rates are observed in Eastern and Central Europe [2]. It has been found that the risk of OC in a woman’s life is 1 in 75, and 1 in 100 cases of this disease is fatal [3,4,5]. It has been estimated that most cases of ovarian cancer are diagnosed in women aged 55–64, and the mean age of deaths due to OC is 70 years old [6]. Since the main causes of the low survival in OC patients are a late diagnosis of the disease due to a lack of symptoms, a lack of effective screening tools and recurrent chemotherapy-resistant disease [7], less than 25% of diagnosed cancers are confined to the ovary only, and most patients have metastatic disease. Further worsening the situation, the lack of specific symptoms indicating the disease means that only 25% of OC cases are diagnosed in stage I [6]. The 5-year survival rate in stages I and II (early disease) is about 90%, while in stages III and IV (advanced disease), it is only 20–40% [8]. This underlines the great need to develop effective biological markers that can detect advanced disease rapidly, which, in turn, will increase the effectiveness of treatments and improve patient survival.

Generally, ovarian cancer is a fairly diverse group of cancers occurring at different locations in the peritoneal cavity. The most common form of OC is an epithelial tumor that develops from superficial ovarian epithelial cells and can evolve into two histological types. Type I, mucous serous low-grade and clear-cell carcinomas have a slower disease progression and account for only 10% of OC deaths. Type II accounts for 60–80% of all ovarian epithelial cancers and includes high-grade serous and undifferentiated sarcomas and carcinosarcomas, which manifest with a much more aggressive tumor biology. It should be mentioned that type II tumors are most often diagnosed in stage III or IV, where a full recovery is practically impossible [6,9].

The International Federation of Gynecology and Obstetrics (FIGO) has developed a classification system for staging ovarian cancer, which is based mainly on the examination of surgically removed tissue (surgical stage), and, importantly, this method complies with the commonly used TMN system (Tumor, Lymph node and Metastasis) [10]. The FIGO system has been shown to be of great prognostic value, which may explain the fact that the higher degree of advancement reflects the biological aggression of the tumor [11]. It was determined [8] that the presence of malignant pleural effusion in stage IV patients, in the absence of other criteria, is associated with a much worse prognosis than in stage III patients. For this reason, the development of a valuable marker indicating an advanced disease is crucial because a faster diagnosis will enable the immediate introduction of therapies, which will translate into extending the life of the OC patient.

TLRs (Toll-like receptors) are a group of receptors included in the PRRs (pattern recognition receptors), which are a family of membrane proteins expressed on immune and epithelial cells [12]. They are able to recognize specific PAMPs (pathogen-associated molecular patterns) and therefore trigger a cascade of reactions leading to the formation of an immune response. Moreover, some TLRs (TLR2, TLR3 and TLR4) respond to endogenous “stress” proteins such as heat shock proteins (Hsp), hyaluronan and fibrinogen [13]. It has been reported [14] that TLR expression in neoplastic cells can promote the persistence of inflammation and cell survival in the neoplastic tumor microenvironment. TLR2 has been shown to be expressed in the normal ovarian epithelium but also to be involved in the pathogenesis of OC. It has been reported to play a key role in detecting the danger signals released by cancer cells and is also involved in triggering the mechanisms of innate immunity and the specific anti-tumor response. However, TLR2 activity causes an immunosuppressive effect, which promotes tumor progression [13]. On the other hand, studies showed [15] that TLR2 expressed on the surface of epithelial ovarian cancer stem cells plays a role in promoting a pro-inflammatory microenvironment, which supports the process of stem cell repair and self-renewal, leading to tumor recurrence.

In this study, we determined the expression of TLR2 in patients with ovarian cancer at various stages of the disease to demonstrate the utility of this receptor as a biomarker for advanced stage OC, which may help to improve the diagnosis process and thus increase survival in patients.

## 2. Materials and Methods

### 2.1. Patient Characteristics

We recruited 50 prospective patients of the I Chair and Department of Gynecological Oncology and Gynecology of the Medical University of Lublin and Oncology Clinic of the Holycross Cancer Center in Kielce, Poland, with a diagnosis of ovarian cancer. The inclusion criterion for this study was the suspicion of ovarian cancer based on gynecological examination, ultrasound or CT scan. The patients had not been previously treated for ovarian cancer and had not received any chemotherapy and/or immunotherapy. Blood samples were obtained from previously untreated women with suspected ovarian cancer one day before surgery. Only patients who had ovarian cancer confirmed intraoperatively or in the histopathological examination following the surgery were included in the study.

The control group included 30 age-matched healthy women reporting no gynecological or non-gynecological symptoms indicative of endometriosis. Their health status was confirmed via routine diagnostic examinations performed during control visits to a gynecologist.

The exclusion criteria were as follows: taking medications affecting the immune system, hormonal therapy, infection during the last four weeks prior to the study, blood transfusion, autoimmune disease, cancer, allergies, pregnancy or lactation.

The Local Bioethical Committee of the Medical University of Lublin approved the study protocol (approval no. KE-0254/251/2014, 27 September 2018), and all the patients gave written informed consent to use their blood samples in the study. The stage of ovarian cancer was evaluated according to the revised International Federation of Gynecology and Obstetrics (FIGO) 2014 classification [16].

### 2.2. Collection of Peripheral Blood and Peritoneal Fluid Samples

The peripheral blood of 50 previously untreated patients with newly diagnosed OC at various stages of the disease and 30 healthy controls was examined during the research. A total of 12 mL of peripheral blood samples was collected from each participant. Part of the blood sample (10 mL) was placed in EDTA tubes for use in flow cytometry to analyze basic lymphocyte subsets and assess the soluble TLR2 concentration in plasma (S-Monovette; SARSTEDT AG & Co. KG, Nümbrecht, Germany). The remaining 2 mL was placed in heparin tubes for use in flow cytometry to analyze monocytes and DCs (Blood Gas Monovette, Luer, Lithium Heparin calcium-balanced; SARSTEDT AG & Co. KG). The serum levels of the tumor markers (cancer antigen 125 (CA-125) and cancer antigen 19-9 (CA19-9)) were also measured and recorded [17]. Peritoneal fluid (2 mL) was obtained during operation from patients with ovarian cancer.

### 2.3. Flow Cytometry and Sample Preparation: Monocytes and Dendritic Cells

Flow cytometry was used to find out the percentage of monocytes and DCs that express TLR2. The process used for sampling was as follows: for each participant, a whole blood sample of 100 μL and fluorochrome-conjugated monoclonal antibodies underwent incubation in the dark for 20 min against these antigens:□CD1c (BDCA-1) FITC/Pacific Blue anti-Human Lineage Cocktail (anti-CD3, CD14, CD16, CD19, CD20, CD56)/TLR2 PE (Biolegend, San Diego, CA, USA);□BDCA-2 FITC/CD123 Pe-Cy7/CD45 V450/TLR2 PE (Biolegend);□CD14 FITC/CD16 V450/HLA-DR Pe-Cy7 (BD Biosciences, San Jose, CA, USA) and TLR2 PE (Biolegend). 

Thereafter, Lysing Buffer (BD Pharm Lyse) was used to treat the samples, and they were then washed in PBS solution (Sigma-Aldrich, St. Louis, MO, USA). The Cytoflex LX (Beckman Coulter, Brea, CA, USA) was used to collect the samples. The data were analyzed using the Kaluza Analysis software and evaluated with dot plots. The definition of mDCs was BDCA1+ Lin− cells; plasmacytoid DCs, BDCA2+ CD123+ cells; classical monocytes, CD14+ CD16− cells; and non-classical monocytes, CD14+ CD16+. A gating step with HLA-DR was added to improve the monocyte purity and isotype controls (Biolegend) used for the determination of binding that was unspecific.

### 2.4. Flow Cytometry and Sample Preparation: Lymphocytes, Natural Killer and Natural Killer T-like Cells

The percentage of peripheral lymphocytes expressing TLR2 was measured. Blood samples of 50 µL with the following antibodies were used:□CD19 FITC/CD3 PE;□CD4 FITC/CD8 PE/CD3 PerCP;□CD3 FITC/CD16 PE/CD56PE;□TLR 2 PE/CD4FITC;□TLR2PE/CD8FITC;□TLR2PE/CD19FITC (BD Biosciences, San Jose, CA, USA).

Thereafter, the samples were treated with Lysing Buffer (BD Pharm Lyse) and PBS solution (Sigma Aldrich) for washing. We then collected the samples using a Cytoflex LX (Beckman Coulter) and analyzed the data using the Kaluza Analysis program. The data were evaluated with dot plots. Isotype controls (Biolegend) were used to determine unspecific binding.

We also calculated the percentages of the following: T helper cells (CD3+ CD4+), T cytotoxic cells (CD3+ CD8+), B lymphocytes (CD3− CD19+), natural killer cells (CD3− CD16+ CD56+) and natural killer T-like cells (CD3+ CD16+ CD56+). Using forward and lateral dispersions and a two-color fluorescence plot, we established the population of target cells. Gates were placed around individual cell populations to determine the relative CD45+ cell percentages. To determine the background signal and to exclude contamination and cell aggregates, FITC IgG1 κ and PE IgG1 κ were used as isotype controls.

### 2.5. Measurement of TLR2 Concentration in Serum and Peritoneal Fluid

The concentration of soluble TLR2 was measured using the RayBio Human TLR2 ELISA Kit ((sensitivity 0.32 ng/mL) RayBiotech Life, Norcross, GA, USA) according to the manufacturer’s recommendations. The results were measured with an automatic reader, VICTOR3 (Perkin Elmer, Waltham, MA, USA), which measures the absorbance of light in the tested material and compares it with control samples of a known concentration. The WorkOut Software plotted linear curves, and based on these, the concentration of soluble antigen in the samples was calculated.

### 2.6. Statistical Analysis

Tibco Statistica 13.3 (StatSoft, Kraków, Poland) was used for data analysis, and the Shapiro–Wilk test was used to test for a normal distribution of continuous variables. Parametric values were presented as mean and standard deviation (SD) for a normal distribution of data, and median, lowest and highest values for an abnormal distribution. For independent variables, a t test was used together with the Mann–Whitney U test to compare differences in the intergroups. We analyzed the differences between two or more groups using the Kruskal–Wallis test and multiple comparisons of mean ranks (post hoc analysis). Further, to determine how accurate the diagnostic test was, we used receiver operating characteristic (ROC) curves for parameters related to the various categories of participants. We then calculated and conducted a comparison of areas under the ROC curves (AUCs) for each parameter (error, 5%; significance, *p* = <0.05). 

## 3. Results

### 3.1. Characteristics of the Study and the Control Group

The characteristics of the study and the control group are presented in Table 1.

### 3.2. Dendritic Cells, Monocytes and Basic Peripheral Blood Lymphocyte Subsets and Expression of TLR2 Antigen in Patients with Ovarian Cancer and Control Group

Table 2 shows the dendritic cell, monocyte and basic peripheral blood lymphocyte subset counts, as well as the percentage of these cell subsets expressing the TLR2 antigen in the patients and controls. Figure 1 presents those variables in patients with particular stages of ovarian cancer.

### 3.3. Receiver Operating Characteristic (ROC) Curve Analysis to Determine the Diagnostic Accuracy of TLR2 Expression on Myeloid Dendritic Cells, Plasmacytoid Dendritic Cells, Monocytes, T Lymphocytes and B Lymphocytes in Patients with Ovarian Cancer vs. Controls

Table 3 shows the ROC analysis in the differentiation of patients with ovarian cancer. Figure 2 and Figure 3 present biomarkers of excellent diagnostic accuracy.

## 4. Discussion

Ovarian cancer is a global problem that affects women of all ages but is most commonly diagnosed in the postmenopausal period [18,19]. Due to the lack of effective screening and the usually asymptomatic course of the disease in its early stages, more than 75% of affected women are diagnosed too late, resulting in less than half of the patients diagnosed with OC surviving more than five years after their diagnosis [19,20]. It has been observed that a good prognosis for women with OC is directly related to the severity of the disease at the time of diagnosis [21]. Women diagnosed with stage I OC have a 5-year survival rate of 90%. On the other hand, for women with confirmed regional disease, the 5-year survival rate drops to 80%, and in those with confirmed metastases, it drops to 25% [21].

The U.S. Preventive Services Task Force and ESGO do not recommend any method for ovarian cancer screening [22]. In cases where ovarian cancer is suspected, we most often use the RMI (risk of malignancy) index or the ROMA or IOTA ADNEX test. Transvaginal ultrasound, serum cancer antigen 125, pelvic examination and HE4 are methods commonly used in the diagnosis and treatment evaluation of ovarian cancer.

In recent years, about 100 new markers of ovarian cancer have been described, clinically evaluated and compared to CA125. Such markers include: kallikreins 6 and 10, osteopontin, haptoglobin, HE4, claudin 3, VEGF, MUC1, CA19-9 and mesothelin [23]. However, only mesothelin and HE4 have met researchers’ expectations. Rosen et al. [23] conducted studies assessing the expression of the above-mentioned markers in patients with ovarian cancer with a low serum CA125 concentration, in whom tumor cells showed low or no CA125 expression. Only mesothelin and HE4 showed high specificity, and it is clinically important that these markers are upregulated in serum ovarian cancer. This provides hope that they will be used as markers complementing or replacing the previously used CA125 serum level determinations.

The FDA recently approved two new tests to diagnose ovarian cancer in women with pelvic tumors: OVA1 and ROMA.

The OVA1 test involves the determination of five markers in the blood serum: prealbumin, apolipoprotein A1, b2-microglobulin, transferrin and CA125. The use of the OVA1 test shows 96% sensitivity and 28% specificity in postmenopausal women, while in premenopausal women, it shows 85% sensitivity and 40% specificity. The OVA1 test detected 76% of malignant lesions that would not have been diagnosed with CA125 alone [24].

The ROMA (risk of malignancy algorithm) is a mathematical algorithm that combines the diagnostic value of serum CA125 and HE4 determinations in relation to the patient’s menopausal status. ROMA results of >13.1% in premenopausal women and >27.7% in postmenopausal women indicate a high probability of malignancy in the ovary. The test shows 100% sensitivity and 74% specificity for premenopausal women, and 92% sensitivity and 76% specificity for postmenopausal women [25].

Both tests are very useful in the differential diagnosis of pelvic tumors in women, but due to their high cost, they cannot be used as screening tests for asymptomatic women. Therefore, many studies are focused on the search for new potential markers that would be characterized by high sensitivity and specificity and a stable and constant concentration in the blood serum, which could complement or replace the most commonly used diagnostic method thus far.

In the present study, we assessed TLR2 expression in blood samples from ovarian cancer patients with various stages of FIGO (Table 3). We found increased TLR2 expression in OC women compared to the control group.

It is widely known that Toll-like receptors play a key role in activating the immune response. TLRs have been shown to be critical mediators of chronic inflammation in the tumor microenvironment and favor tumor cell survival [26]. TLR signaling plays an important role in different types of cancer, and the role of their expression may vary depending on the type of cancer [27]. Although TLR2 is one of the most widely studied members of the TLR family, there are still no conclusive studies on the impact of TLR2 expression in OC [28].

In the available literature, several studies were found to assess the effect of TLR2 expression in other neoplasms. Li et al. [29] showed in their studies that patients in the high-grade glioma group had a significantly higher TLR2 expression compared to the low-grade glioma group [29]. Recent research presented by Wang et al. [28] in which TLR2 values were assessed for prognosis in breast cancer (BCa) showed that a high TLR2 expression in patients with BCa is associated with poor overall survival compared to patients with low TLR2 levels [28]. The pivotal role of TLRs has also been confirmed in cervical intra-epithelial neoplasia 2, suggesting the role of TLRs in the regression of CIN2, and providing hope for a potential use of TLR agonists in the treatment of this type of lesion [30]. Additionally, research has been performed confirming the role of TLRs in vulvar cancer [31]. An interesting issue is the single-nucleotide polymorphisms (SNPs) in TLR-coding genes, which may potentially serve as markers for early cancer susceptibility [32]. In a study concerning cervical cancer, it was registered that TLR4 haplotype GTAC and TLR9 haplotype GATC are associated with an increased risk of cervical cancer, while TLR4 haplotype GCAG is associated with a decreased risk [32]. Such genetic observation is still ahead of the research concerning ovarian cancer and might be the next step after confirming the role of TLRs in this type of cancer. The potency of TLR2 in HPV-associated cervical tumors has also been observed in the study of Zom et al. [33], who conjugated HPV16-encoded synthetic long peptides (SLPs) and the TLR2 ligand and registered that the conjugated SLPs were efficiently processed by antigen-presenting cells (APCs) and the immunological response was triggered. This is analogical to the elevated level of TLRs obtained by us, but TLR9 has been observed in patients with cervical neoplasia [34], also stating that the TLR4 and TLR9 expressions may be upregulated by HPV16 infection [35].

The studies of Vlad et al. [36] showed that patients with early-stage ovarian cancer (FIGO I–III) and TLR9 positivity showed statistically significantly higher survival compared to patients without expression. Whether or not the elevated expression is linked to any infection is not fully understood, but with no doubt, TLR2 may be a potential candidate to serve as a biomarker of ovarian cancer, especially in the early stage.

## 5. Conclusions

The role of TLRs in pathological conditions caused by both infectious and non-infectious agents is a known scientific fact, yet, to our best knowledge, this is the first study to identify the role of TLR2 expression as a prognostic factor in patients with OC. It occurred in the present study that patients with OC are characterized with an elevated level of TLR2 in comparison to healthy individuals, suggesting that TLR2 may serve as a promising non-invasive biomarker of ovarian cancer. Further studies are needed to check the ethological background of this state.

## Figures and Tables

**Figure 1 biomolecules-11-01205-f001:**
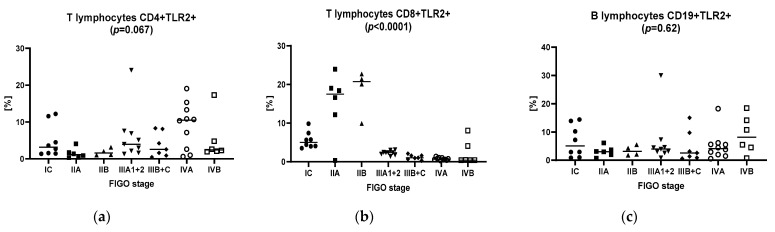
The frequencies of the dendritic cell, monocyte and basic peripheral blood lymphocyte subsets expressing the TLR2 antigen of ovarian cancer patients with different FIGO stages. (**a**) T lymphocytes CD4+ TLR2+; (**b**) T lymphocytes CD8+ TLR2+; (**c**) B lymphocytes CD19+ TLR2+; (**d**) myeloid dendritic cells BDCA1+ CD19− TLR2+; (**e**) plasmacytoid dendritic cells BDCA2+ CD123+ TLR2+; (**f**) classical monocytes CD14+ CD16− TLR2+; (**g**) non-classical monocytes CD14+ CD16+ TLR2+; (**h**) TLR2 concentration in peritoneal fluid; (**i**) TLR2 concentration in serum.

**Figure 2 biomolecules-11-01205-f002:**
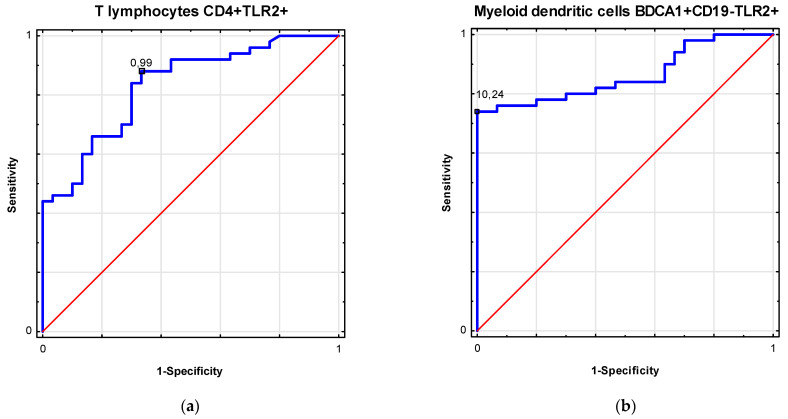
Receiver operating curve (ROC) analysis to determine diagnostic accuracy in differentiation of patients with ovarian cancer: (**a**) frequencies of T lymphocytes CD4+ TLR2+ (%); (**b**) frequencies of myeloid dendritic cells BDCA1+ CD19− TLR2+ (%).

**Figure 3 biomolecules-11-01205-f003:**
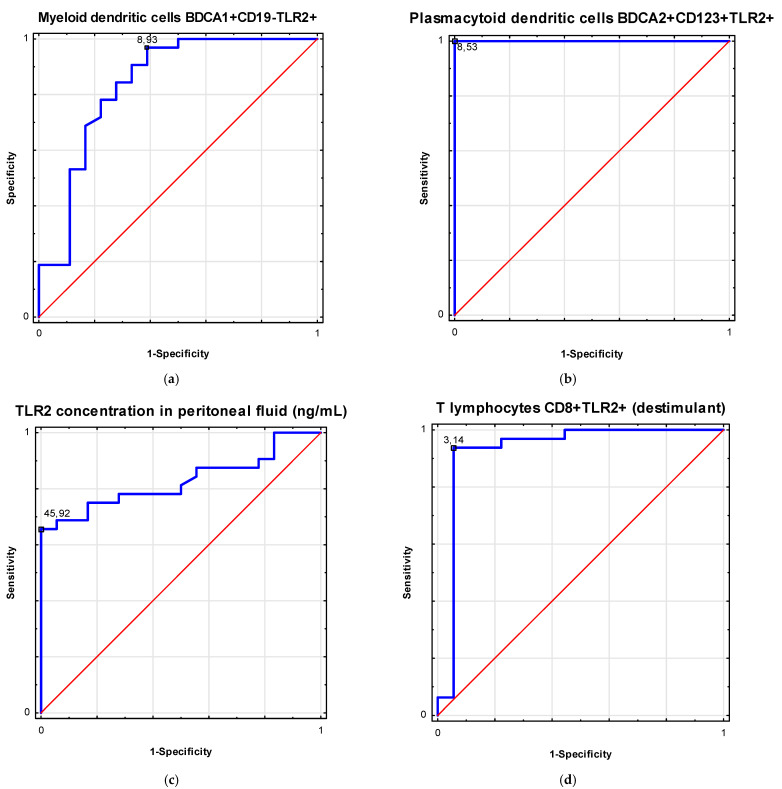
Receiver operating curve (ROC) analysis to determine diagnostic accuracy in differentiation of patients with ovarian cancer in III/IV FIGO stages and FIGO stages I/II: (**a**) frequencies of myeloid dendritic cells BDCA1+ CD19− TLR2+ (%); (**b**) frequencies of plasmacytoid dendritic cells BDCA2+ CD123+ TLR2+ (%); (**c**) TLR2 concentration in peritoneal fluid (ng/mL); (**d**) frequencies of T lymphocytes CD8+ TLR2+ (%) as a destimulant.

**Table 1 biomolecules-11-01205-t001:** Basic characteristics of the study and the control groups.

Parameter	Ovarian Cancer (*n* = 50)		Control Group (*n* = 30)	
	Mean ± SD/Median (Range)		Mean ± SD/Median (Range)	
Age (years)	54.1 ± 8.0		56.4 ± 10.1	
Ca-125 (U/mL)	92.3 ± 45.9		13.8 ± 7.2	
CA 19-9 (U/mL)	114.8 ± 55.8		17.3 ± 5.1	
Pregnancies	1.0 (0–5)		2.0 (0–7)	
BMI	27.6 (18–39)		25.0 (18–36)	
**FIGO stage**	**N**	**%**	**N**	**%**
IC	8	16	N/A	
IIA	6	12		
IIB	4	8		
IIIA1	7	14		
IIIA2	2	4		
IIIB	2	4		
IIIC	5	10		
IVA	10	20		
IVB	6	12		
**Histotypes**	**N**	**%**	**N**	**%**
Low-grade serous carcinoma	15	N/A	N/A	
Low-grade mucinous carcinoma	3			
Low-grade endometrioid carcinoma	5			
Undifferentiated carcinoma	23			
Carcinosarcoma	2			
Granulosa cell tumor	1			

N/A = not applicable, CA-125 = cancer antigen 125, CA19-9 = cancer antigen 19-9.

**Table 2 biomolecules-11-01205-t002:** Dendritic cell, monocyte and basic peripheral blood lymphocyte subset counts, as well as the frequencies of these cell subsets expressing the TLR2 antigen in the study and control group.

Parameter [%]	Ovarian Cancer Group (*n* = 50)	Healthy Control Group (*n* = 30)	t/Z Value	*p*-Value
	Mean ± SD/Median (Range)			
White blood cells (10^3^/mm^3^)	8.2 ± 1.7	7.3 ± 0.65	2.8	0.0057
Neutrophils (10^3^/mm^3^)	4.9 ± 1.5	4.26 ± 0.9	2.1	0.037
Monocytes (10^3^/mm^3^)	0.51 ± 0.17	0.44 ± 0.12	1.8	0.069
Lymphocytes (10^3^/mm^3^)	2.3 ± 0.7	2.5 ± 0.5	−1.5	0.13
Myeloid dendritic cells BDCA1+ CD19−	0.26 (0.04–0.72)	0.42 (0.12–0.69)	−2.5	0.012
Plasmacytoid dendritic cells BDCA2+ CD123+	0.33 (0.14–0.99)	0.26 (0.07–0.58)	2.4	0.016
Myeloid dendritic cells BDCA1+ CD19−/Plasmacytoid dendritic cells BDCA2+ CD123+ ratio	0.74 (0.11–4.4)	1.6 (0.37–5.3)	−3.5	0.0004
Classical monocytes CD14+ CD16−	87.0 ± 5.4	91.1 ± 3.3	−4.3	<0.0001
Non-classical monocytes CD14+ CD16+	8.45 (2.3–20.5)	4.6 (2.2–13.5)	3.8	0.0001
T lymphocytes CD3+	72.9 (62.0–79.9)	71.5 (63.0–78.2)	0.85	0.39
B lymphocytes CD19+	11.0 (6.7–17.1)	11.8 (6.5–17.0)	−0.97	0.33
NK cells CD3− CD16+ CD56+	11.8 ± 4.3	14.6 ± 3.3	−3.1	0.0024
NKT-like cells CD3+ CD16+ CD56+	2.2 (0.2–10.1)	3.3 (1.2–5.0)	−1.6	0.099
T lymphocytes CD3+ CD4+	40.7 ± 5.7	40.4 ± 3.3	0.25	0.8
T lymphocytes CD3+ CD8+	29.5 ± 5.5	30.4 ± 3.8	−0.8	0.4
T lymphocytes ratio CD3+ CD4+/T CD3+ CD8+	1.35 (0.73–3.0)	1.3 (0.96–1.96)	0.48	0.63
Myeloid dendritic cells BDCA1+ CD19− TLR2+	17.1 (1.6–52.0)	4.1 (1.25–10.0)	−2.5	0.012
Plasmacytoid dendritic cells BDCA2+ CD123+ TLR2+	10.6 (2.5–43.9)	5.4 (1.0–16.5)	3.7	0.0002
Classical monocytes CD14+ CD16− TLR2+	5.4 (2.3–30.3)	4.0 (1.95–16.0)	2.3	0.021
Non-classical monocytes CD14+ CD16+ TLR2+	10.0 (1.5–35.8)	4.7 (0.9–18.0)	3.2	0.0014
T lymphocytes CD4+ TLR2+	2.7 (0.3–24.1)	0.82 (0.08–3.5)	4.9	<0.0001
T lymphocytes CD8+ TLR2+	2.2 (0.3–24.0)	1.1 (0.19–6.0)	2.5	0.013
B lymphocytes CD19+ TLR2+	3.8 (0.6–30.0)	2.5 (0.17–6.9)	2.5	0.012
TLR2 concentration in serum (ng/mL)	11.5 (2.5–74.9)	3.6 (0.34–20.9)	4.4	<0.0001

**Table 3 biomolecules-11-01205-t003:** Receiver operating curve (ROC) analysis to determine diagnostic accuracy in differentiation of patients with ovarian cancer.

Factor	Parameter (%)	Prognostic Value	Youden Index	Area under the Curve (AUC)	95% CI	*p*-Value
Ovarian cancer	T lymphocytes CD4+ TLR2+	0.99	0.55	0.83	0.74–0.92	<0.0001
	Myeloid dendritic cells BDCA1+ CD19− TLR2+	10.24	0.74	0.86	0.78–0.94	<0.0001
FIGO Stages III–IV	Myeloid dendritic cells BDCA1+ CD19− TLR2+	8.93	0.58	0.84	0.71–0.97	<0.0001
	Plasmacytoid dendritic cells BDCA2+ CD123+ TLR2+	8.53	1.0	1.0	1.0	<0.0001
	TLR2 concentration in peritoneal fluid (ng/mL)	45.92	0.66	0.83	0.71–0.94	<0.0001
	T lymphocytes CD8+TLR2+ (destimulant)	3.14	0.88	0.93	0.83–1.0	<0.0001

## Data Availability

Due to privacy and ethical concerns, the data that support the findings of this study are available on request from the first author (M.S.).

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
