# Peer review of "Toll-like Receptor 2 as a Marker Molecule of Advanced Ovarian Cancer"

_biomolecules, 2021, doi:10.3390/biom11081205_

Round 1
Reviewer 1 Report
Please make the text of the Results Section precisely describe the findings, in addition to the tables. While there is a presentation of Type I & Type II ovarian cancers, the paper should express the data in terms of these discriminators. Accordingly, describe relationships to grade. Discuss strengths and limitations. It is extremely important to document the sub-types of ovarian cancer that are included. Some questions about the inclusion/exclusion of LMPs should be answered.
I have highlighted spots in the manuscript that I feel the text should be improved.

Author Response
Dear Reviewer,
Thank you for giving the opportunity to submit a revised draft of our manuscript titled: “Toll-like receptor 2 as a marker molecule of advanced ovarian cancer“ to Biomolecules. We appreciate the time and effort that you have dedicated to providing your valuable feedback on our manuscript. We have been able to incorporate changes to reflect most of the suggestions provided by you. We have highlighted the changes within the manuscript.
Here is a point-by-point response to the reviewers’ comments and concerns.
Reviewer 1
Please make the text of the Results Section precisely describe the findings, in addition to the tables. While there is a presentation of Type I & Type II ovarian cancers, the paper should express the data in terms of these discriminators. Accordingly, describe relationships to grade. Discuss strengths and limitations. It is extremely important to document the subtypes of ovarian cancer that are included. Some questions about the inclusion/exclusion of LMPs should be answered. I have highlighted spots in the manuscript that I feel the text should be improved.
RE: Thank you very much for your insightful review. The parts marked in yellow have been corrected / clarified and marked in red.
Again, we would like to thank you for your time and consideration devoted into the corrections and suggestions of our paper. We hope that in the corrected form the paper will fulfill the requirements.
Thank you,
On behalf of the Authors,
Paulina Niedźwiedzka-Rystwej
Reviewer 2 Report
The authors examined the efficacy of Toll-like receptor 2 as a marker of ovarian cancer screening. The results of this study are of some interest. However, current evidence shows that routine screening for ovarian cancer is not recommended. The reviewer has several concerns as follows.
Lines 38-40
The authors should use the latest version of cancer statistics.
Lines 45-49
Suggest rephrasing for a better understanding.
Lines 55-82
These sentences are abundant and need to be revised.
Materials and Methods
Inclusion criteria of patients are lacking.
Table 1
What is pregnancies?
Histopathological subtypes were not determined.
BMI should be matched between the study group and the control group since higher BMI appears to be associated with higher expression of TLR2.
Figure 1
The authors may add the value of the control group.
Lines 279-281
This sentence is difficult to read. Suggest rephrasing.
Discussion
The U.S. Preventive Services Task Force does not recommend routine screening for ovarian cancer using any method. Transvaginal ultrasound and serum cancer antigen 125 testings are readily available procedures that are commonly used to evaluate women with signs or symptoms of ovarian cancer and have been evaluated in screening studies. Pelvic examination is also commonly performed to evaluate women with lower abdominal symptoms. The authors should discuss the role of ovarian cancer screening.
Overall, the reviewer feels that the authors should focus on women with early-stage of ovarian cancer to examine the efficacy of TLR-2 as a marker of ovarian cancer screening. The authors need to discuss ovarian cancer screening deeply.
Author Response
Dear Reviewer,
Thank you for giving the opportunity to submit a revised draft of our manuscript titled: “Toll-like receptor 2 as a marker molecule of advanced ovarian cancer“ to Biomolecules. We appreciate the time and effort that you and the reviewers have dedicated to providing your valuable feedback on the manuscript. We have been able to incorporate changes to reflect most of the suggestions provided by the reviewers. We have highlighted the changes within the manuscript.
Here is a point-by-point response to the reviewer’ comments and concerns:
Reviewer 2
The authors examined the efficacy of Toll-like receptor 2 as a marker of ovarian cancer screening. The results of this study are of some interest. However, current evidence shows that routine screening for ovarian cancer is not recommended. The reviewer has several concerns as follows.
Lines 38-40
The authors should use the latest version of cancer statistics.
RE: The text includes the latest statistics on ovarian cancer.
Lines 45-49
Suggest rephrasing for a better understanding.
RE: These lines have been rephrased as suggested.
Lines 55-82
These sentences are abundant and need to be revised.
RE: Revised as suggested.
Materials and Methods
Inclusion criteria of patients are lacking.
RE: The inclusion criteria have been completed.
Table 1
What is pregnancies?
RE: Pregnancies are the number of pregnancies of the patients have had in the past.
Histopathological subtypes were not determined.
RE: Histopathological subtypes are determined in Table 1.
BMI should be matched between the study group and the control group since higher BMI appears to be associated with higher expression of TLR2.
RE: The reported differences in BMI were not statistically significant. We chose a control group, which consisted of 30 people perfectly matched with BMI for 30 patients, the remaining 20 people that we wanted to include in the control group turned out to not meet the inclusion criterion for the study due to comorbidities.
Figure 1
The authors may add the value of the control group.
RE: We did not add the control group due to the lack of clarity, and all the data is in Table 2, but if the Reviewer decides that this is needed, we will gladly correct it.
Lines 279-281
This sentence is difficult to read. Suggest rephrasing.
RE: We rephrased the sentence.
Discussion
The U.S. Preventive Services Task Force does not recommend routine screening for ovarian cancer using any method. Transvaginal ultrasound and serum cancer antigen 125 testings are readily available procedures that are commonly used to evaluate women with signs or symptoms of ovarian cancer and have been evaluated in screening studies. Pelvic examination is also commonly performed to evaluate women with lower abdominal symptoms. The authors should discuss the role of ovarian cancer screening.
Overall, the reviewer feels that the authors should focus on women with early-stage of ovarian cancer to examine the efficacy of TLR-2 as a marker of ovarian cancer screening. The authors need to discuss ovarian cancer screening deeply.
RE: We tried to correct to our best knowledge and possibilities.
Again, we would like to thank you again for your time and consideration devoted into the corrections and suggestions of our paper. We hope that in the corrected form the paper will fulfill the requirements.
Thank you,
On behalf of the Authors,
Paulina Niedźwiedzka-Rystwej
Round 2
Reviewer 2 Report
Since the authors did good job to revise the manuscript, I have no extra comments to the authors.